# 6D Virtual Sensor for Wrench Estimation in Robotized Interaction Tasks Exploiting Extended Kalman Filter

**Loris Roveda [1,*](ID), Andrea Bussolan [2], Francesco Braghin [2] and Dario Piga [1]**

[1] Istituto Dalle Molle di studi sull'Intelligenza Artificiale (IDSIA), Scuola Universitaria Professionale della Svizzera Italiana (SUPSI), Università della Svizzera Italiana (USI), 6928 Manno, Switzerland; dario.piga@supsi.ch

[2] Department of Mechanical Engineering, Politecnico di Milano, 20156 Milano, Italy; andrea.bussolan@mail.polimi.it (A.B.); francesco.braghin@polimi.it (F.B.)

* Correspondence: loris.roveda@idsia.ch

**Abstract:** Industrial robots are commonly used to perform interaction tasks (such as assemblies or polishing), requiring the robot to be in contact with the surrounding environment. Such environments are (partially) unknown to the robot controller. Therefore, there is the need to implement interaction controllers capable of suitably reacting to the established contacts. Although standard force controllers require force/torque measurements to close the loop, most of the industrial manipulators do not have installed force/torque sensor(s). In addition, the integration of external sensors results in additional costs and implementation effort, not affordable in many contexts/applications. To extend the use of compliant controllers to sensorless interaction control, a model-based methodology is presented in this paper for the online estimation of the interaction wrench, implementing a 6D virtual sensor. Relying on sensorless Cartesian impedance control, an Extended Kalman Filter (EKF) is proposed for the interaction wrench estimation. The described approach has been validated in simulations, taking into account four different scenarios. In addition, experimental validation has been performed employing a Franka EMIKA panda robot. A human–robot interaction scenario and an assembly task have been considered to show the capabilities of the developed EKF, which is able to perform the estimation with high bandwidth, achieving convergence with limited errors.

**Keywords:** extended kalman filter; wrench estimation; 6D virtual sensor; sensorless cartesian impedance control; industrial robots; interaction robotized tasks

---

## 1. Introduction

### 1.1. Context

Robots are increasingly involved in daily life activities, which no longer consist of only repetitive simple tasks, but rather require interaction with an ever-changing environment, while performing a multitude of different tasks [1,2]. Considering the manufacturing context, the efforts that have to be deployed to pre-program all the possible tasks and scenarios are excessive; therefore, robots have to provide a flexible solution, adapting to new tasks/production while guaranteeing target performance [3,4]. In these complex scenarios the robot is required to learn and suitably modify its behavior on the basis of the operating conditions [5] and considering interaction tasks (i.e., robot exchanging forces/torques with the environment) the capability to adapt becomes even more critical [6]. To avoid any unwanted/unstable behavior, the interaction force must be controlled [7,8]. Common interaction control strategies, however, make use of expensive sensors [9–11], increasing hardware

costs and implementation efforts not affordable in many contexts/applications. To avoid the use of such devices, with the robot able to adapt to uncertain interaction, many works are investigating external wrench estimation algorithms and sensorless control methodologies.

*1.2. Related Works*

The research community pays much attention to the achievement of a stable interaction between sensorless robots and the environment, employing efforts in developing force–sensorless methodologies to estimate the interaction between the robot and its environment. Such research area is strongly connected with the main aim of this paper, since the proper estimation of the established interaction between the robot and its environment is required to design a stable and high-performance interaction controller for a sensorless robot. Some approaches [12] focus on the derivation of high-accuracy models which are then used to estimate the interaction wrench during task execution. In the literature, to prevent the use of force sensors, disturbance observers-based solutions are developed, where the external wrench applied to the manipulator is observed exploiting the inverse of the robot model. In [13] a nonlinear disturbance observer is proposed to estimate the external interaction, allowing for the first time the possibility to consider the intrinsic nonlinearities of systems such as robot manipulators. This method guarantees the stability of the disturbance observer by properly tuning its design parameters while taking into account the physical parameters and constraints (e.g., maximum joint velocities) of the robot. A more general approach is developed in [14] where a disturbance state observer is coupled with a machine learning techniques, which are implement to identify a task-oriented dynamic model. The use of learning-based approach prevents the modeling of the robot dynamics terms (such as joints' friction or Coriolis effects). To eliminate modeling errors, in [15] a parametric dynamics robot model coupled with machine learning techniques is designed. In this work a two-layer modeling is implemented with the combination of rigid-body dynamics (RBD) and a compensator trained with multilayer perception (MLP), ensuring a better model accuracy than each of the two model taken individually. To perform the interaction force estimation, such a modeled dynamics is then exploited in a disturbance Kalman filter based on a time-invariant composite robot model, providing robust estimation against uncertainty. To avoid the use of acceleration measurements and the computation of the inverse of the robot mass matrix that amplifies measurements noise, in [16] a sensorless admittance control scheme exploiting a generalized momentum-based disturbance observer to model a linear environment dynamics is proposed. A radial basis neural networks approach (RBNN) is used to compensate model uncertainties and, in order to properly manage the control inputs, actuation saturation is considered. Some state-of-the-art works structure the identification as an optimization problem. A different methodology that shows a deep connection with the generalized momentum-based approach, highlighted by simulation and experimental results, is developed in [17] where the filtered dynamic equations are combined with a recursive least-square estimation algorithm to provide a smooth external force estimation. Friction models, such as Coulomb friction, show uncertainties connected with the joint velocity, especially when velocities are close to zero. Solving in real time a convex optimization problem, the method proposed in [18] estimates the reaction force taking into account the aforementioned Coulomb friction uncertainties. Different types of virtual sensors based on high-performance dynamic model calibration have been proposed. A task-oriented calibrated robot dynamic model, which also includes the thermal state of the robot manipulator, is designed in [19]. The proposed dynamic model is calibrated by means of a two-stage optimization which provides suitable paths later combined in exciting trajectories. The estimation of the external force is obtained, using the residual method, as difference between the modeled and measured torques. In other works, Artificial Intelligence has been deployed to map the interaction between the robot and the environment. The designed controller is then based on the learned dynamics. Considering the scenario of working with soft tissue, in [20] the interaction is modeled as a visco-elastic system and the design of the force observer used to estimate the interaction force is based on Lyapunov time-varying equation. The force estimation is then used to develop a robot position controller.

Considering a bilateral tele-operation system, in order to estimate the interaction forces between the slave manipulator and the environment, in [21] an online sparse Gaussian process regression (OSGPR) approach is proposed. The described work does not need any previous knowledge of the slave manipulator dynamic model and it avoids the use of the inverse of Jacobian transpose, but the generalized model is obtained through offline training with previously acquired dataset. The interaction force is obtained in real time by means of the design estimator. Other external sensors can be used instead of force ones to acquire more data useful to assess the interaction force. In [22] exteroceptive sensing (i.e., a depth camera) is used for the detection of contacts while the residual method is deployed to evaluate the external joint torques. This approach provides a reliable estimation of the exchanged force at the contact point even in the scenario of multiple contact points. Optical Coherence Tomography (OCT) images are classified with a Neural Network in [23]. The network is trained on images from a Finite Element Method (FEM) simulation of the deformed sclera, while a Bayesian filter is used to parameterize the model. In [24] Convolutional Neural Networks and Long-Short Term Memory networks are used to process the spatio-temporal information included in video sequences and tool data to assess the interaction force.

*1.3. Paper Contribution*

Extending the work in [25], a model-based methodology is presented in this paper for the online estimation of the interaction wrench, implementing a 6D virtual sensor. Relying on sensorless Cartesian impedance control (to give to the controlled robot a compliant behavior while interacting with an unknown environment), an Extended Kalman Filter (EKF) is proposed for the interaction wrench estimation. The proposed EKF is capable of estimating the forces and torques acting at the robot end-effector, making possible to implement a 6D virtual sensor.

The interaction wrench can be considered to be a deterministic variable (i.e., a model of the interaction between the robot and the environment can be derived and exploited for its estimation). Although other approaches can be used to model the interaction wrench dynamics (such as sequential Monte Carlo, unscented Kalman filter, and particle filtering methodologies [26–30]), they require the measurement (i.e., samples) of the interaction wrench for the training of the algorithm. In many practical cases, this is not possible or, if a force/torque sensor is available, the sensor is also used online, i.e., not requiring the implementation of an estimation algorithm. The here presented approach, instead, exploiting the well-known robot dynamics modeling, is capable of performing the estimation of the interaction wrench without any use of wrench data for the algorithm training.

The described approach has been validated in simulations, taking into account four different scenarios. In addition, experimental validation has been performed employing a Franka EMIKA panda robot. A human–robot interaction scenario and an assembly task have been considered to show the capabilities of the developed EKF, which is able to perform the estimation with high bandwidth, achieving convergence with limited errors.

## 2. Sensorless Cartesian Impedance Control

The Cartesian impedance controller guarantees a compliant behavior during interaction. Such controller must be implemented to design the proposed Extended Kalman Filter. The following manipulator dynamics is considered [31]:

$$\mathbf{B}(\mathbf{q})\ddot{\mathbf{q}} + \mathbf{C}(\mathbf{q},\dot{\mathbf{q}}) + \mathbf{g}(\mathbf{q}) + \boldsymbol{\tau}_f(\dot{\mathbf{q}}) = \boldsymbol{\tau} - \mathbf{J}(\mathbf{q})^T\mathbf{h}_{ext}, \tag{1}$$

where $\mathbf{q}$ is the joint position vector, $\mathbf{B}(\mathbf{q})$ is the inertia matrix, $\mathbf{C}(\mathbf{q},\dot{\mathbf{q}})$ is the Coriolis vector, $\mathbf{g}(\mathbf{q})$ is the gravitational vector, $\boldsymbol{\tau}_f(\dot{\mathbf{q}})$ is the joint friction vector, $\mathbf{J}(\mathbf{q})$ is the Jacobian matrix, and $\mathbf{h}_{ext} = [\mathbf{f},\mathbf{C}]^T$ is the external force/torque vector, $\boldsymbol{\tau}$ is the joint torque vector. Based on Equation (1), the sensorless

Cartesian impedance controller with dynamic compensation [31] is designed, defining the joint torque vector $\boldsymbol{\tau}$ as:

$$\boldsymbol{\tau} = \mathbf{B}(\mathbf{q})\boldsymbol{\gamma} + \mathbf{C}(\mathbf{q}, \dot{\mathbf{q}}) + \mathbf{g}(\mathbf{q}) + \boldsymbol{\tau}_f(\dot{\mathbf{q}}), \tag{2}$$

where $\boldsymbol{\gamma}$ is the sensorless Cartesian impedance control law. Translational $\ddot{\mathbf{p}}$ and rotational $\ddot{\boldsymbol{\varphi}}_{cd}$ (described by the intrinsic Euler angles representation) accelerations of the sensorless Cartesian impedance controller $\boldsymbol{\gamma}$ can be written as:

$$\begin{aligned}
\ddot{\mathbf{p}} &= \mathbf{M}_t^{-1}\left(-\mathbf{D}_t\dot{\mathbf{p}} - \mathbf{K}_t\,\Delta\mathbf{p}\right), \\
\ddot{\boldsymbol{\varphi}}_{cd} &= \mathbf{M}_r^{-1}\left(-\mathbf{D}_r\dot{\boldsymbol{\varphi}}_{cd} - \mathbf{K}_r\,\boldsymbol{\varphi}_{cd}\right).
\end{aligned} \tag{3}$$

These equations describe, respectively, the translational part and the rotational part of the sensorless Cartesian impedance control. Regarding the translational part, $\mathbf{M}_t$ represents the mass matrix, $\mathbf{D}_t$ the damping matrix, $\mathbf{K}_t$ the stiffness matrix, the actual Cartesian positions vector and the target position vector are, respectively, $\mathbf{p}$ and $\mathbf{p}^d$, while $\Delta\mathbf{p} = \mathbf{p} - \mathbf{p}^d$. Instead, concerning the rotational part, $\mathbf{M}_r$ represents the inertia matrix, $\mathbf{D}_r$ and $\mathbf{K}_r$ are again the damping matrix and the stiffness matrix respectively, $\mathbf{R}_c$ is the compliant frame at the end-effector, $\mathbf{R}_d$ is the target frame, and the mutual orientation between these two frames is represented by $\mathbf{R}_c^d = \mathbf{R}_d^T\mathbf{R}_c$, from which the set of Euler angles $\boldsymbol{\varphi}_{cd}$ is extracted. Considering the rotational part of the sensorless Cartesian, it is possible to compute the angular accelerations $\dot{\boldsymbol{\omega}}cd$:

$$\dot{\boldsymbol{\omega}}_{cd} = \mathbf{T}(\boldsymbol{\varphi}_{cd})\left(\mathbf{M}_r^{-1}\left(-\mathbf{D}_r\dot{\boldsymbol{\varphi}}_{cd} - \mathbf{K}_r\boldsymbol{\varphi}_{cd}\right)\right) + \dot{\mathbf{T}}(\boldsymbol{\varphi}_{cd})\dot{\boldsymbol{\varphi}}_{cd}, \tag{4}$$

where matrix $\mathbf{T}(\boldsymbol{\varphi}_{cd})$ defines the transformation from Euler angles derivatives to angular velocities $\boldsymbol{\omega}_{cd} = \mathbf{T}(\boldsymbol{\varphi}_{cd})\dot{\boldsymbol{\varphi}}_{cd}$, and $\boldsymbol{\omega} = \mathbf{R}_{ee}\boldsymbol{\omega}_{cd}$ (with $\mathbf{R}_{ee}$ the rotation matrix from the robot base to its end-effector) [31]. By defining $\widetilde{\mathbf{M}}_r = (\mathbf{R}_{ee}\mathbf{T}(\boldsymbol{\varphi}_{cd}))^{-1}\mathbf{M}_r$ and $\widetilde{\mathbf{D}}_r = \mathbf{D}_r - \widetilde{\mathbf{M}}_r\mathbf{R}_{ee}\dot{\mathbf{T}}(\boldsymbol{\varphi}_{cd})$, Equation (4) can be written as:

$$\dot{\boldsymbol{\omega}} = \widetilde{\mathbf{M}}_r^{-1}\left(-\widetilde{\mathbf{D}}_r\dot{\boldsymbol{\varphi}}_{cd} - \mathbf{K}_r\boldsymbol{\varphi}_{cd}\right). \tag{5}$$

The formulation resulting from Equations (3)–(5), can be written in a compact form as follows:

$$\ddot{\mathbf{x}}^{imp} = -\mathbf{M}^{-1}\left(\mathbf{D}\dot{\mathbf{x}} + \mathbf{K}\,\Delta\mathbf{x}\right), \tag{6}$$

where $\ddot{\mathbf{x}}^{imp} = [\ddot{\mathbf{x}}_t; \ddot{\mathbf{x}}_r] = [\ddot{\mathbf{p}}; \dot{\boldsymbol{\omega}}]$ is the target acceleration computed by the sensorless Cartesian impedance control. $\mathbf{M} = [\mathbf{M}_t\,0; 0\,\widetilde{\mathbf{M}}_r]$, $\mathbf{D} = [\mathbf{D}_t\,0; 0\,\widetilde{\mathbf{D}}_r]$, $\mathbf{K} = [\mathbf{K}_t\,0; 0\,\mathbf{K}_r]$ are the sensorless Cartesian impedance mass, damping and stiffness matrices composed by both the translational and rotational parts, and $\Delta\mathbf{x} = \mathbf{x} - \mathbf{x}^d = [\Delta\mathbf{p}; \boldsymbol{\varphi}_{cd}]$. $\mathbf{x}$ is the current robot end-effector pose vector including both translational and rotational components, while $\mathbf{x}^d$ is the reference robot end-effector pose vector including both translational and rotational components. The sensorless Cartesian impedance control law $\boldsymbol{\gamma}$ can then be written as follows:

$$\boldsymbol{\gamma} = \mathbf{J}(\mathbf{q})^{-1}\left(\ddot{\mathbf{x}}^{imp} - \dot{\mathbf{J}}(\mathbf{q}, \dot{\mathbf{q}})\dot{\mathbf{q}}\right). \tag{7}$$

In general, matrix $\mathbf{J}(\mathbf{q})^{-1}$ can be substituted with the pseudo-inverse of the Jacobian matrix $\mathbf{J}(\mathbf{q})^{\#}$ [32]. Substituting Equation (2) in (1), under the hypothesis that the manipulator dynamics is known (such identification can be performed with state-of-the-art techniques [33]), the controlled robot dynamics results in:

$$\ddot{\mathbf{q}} = \boldsymbol{\gamma} - \mathbf{B}(\mathbf{q})^{-1}\mathbf{J}(\mathbf{q})^T\mathbf{h}_{ext}, \tag{8}$$

where $\mathbf{h}_{ext} = [\mathbf{f}, \mathbf{T}^T(\boldsymbol{\varphi}_{cd})\boldsymbol{\mu}^d]$ (considering the external forces $\mathbf{f}$ and torques $\boldsymbol{\mu}^d$ - referred to the target frame $\mathbf{R}_d$—acting on the robot related to the interaction with the environment). The substitution of Equation (7) into (8) leads to:

$$\mathbf{J}(\mathbf{q})\ddot{\mathbf{q}} + \dot{\mathbf{J}}(\mathbf{q}, \dot{\mathbf{q}})\dot{\mathbf{q}} = \ddot{\mathbf{x}} = \ddot{\mathbf{x}}^{imp} - \mathbf{J}(\mathbf{q})\mathbf{B}(\mathbf{q})^{-1}\mathbf{J}(\mathbf{q})^T\mathbf{h}_{ext},\tag{9}$$

with $\ddot{\mathbf{x}} = \mathbf{J}(\mathbf{q})\ddot{\mathbf{q}} + \dot{\mathbf{J}}(\mathbf{q}, \dot{\mathbf{q}})\dot{\mathbf{q}}$ the resulting Cartesian acceleration of the robot end-effector resulting from the implementation of the proposed sensorless Cartesian impedance controller. Finally, substituting Equation (6) into (9), the controlled robot dynamics resulting from the design of the sensorless Cartesian impedance control is described by the following equation:

$$\mathbf{M}\ddot{\mathbf{x}} + \mathbf{D}\dot{\mathbf{x}} + \mathbf{K}\Delta\mathbf{x} = -\overline{\mathbf{L}}(\mathbf{q})\mathbf{h}_{ext},\tag{10}$$

where $\overline{\mathbf{L}}(\mathbf{q}) = \mathbf{M}\mathbf{J}(\mathbf{q})\mathbf{B}(\mathbf{q})^{-1}\mathbf{J}(\mathbf{q})^T$. The resulting dynamic equation is therefore coupled in the Cartesian degrees of freedom (DoFs) by the matrix $\overline{\mathbf{L}}(\mathbf{q})$.

**Remark 1.** *The sensorless Cartesian impedance control is therefore resulting in a coupled controlled robot dynamics. Matrix $\overline{\mathbf{L}}(\mathbf{q})$ redistributes interaction forces along all the Cartesian DoFs. Although the decoupled robot behavior cannot be achieved implementing such controller, the sensorless Cartesian impedance control strategy allows implementation of a tunable compliant robot behavior, ensuring a safe and stable interaction with the target environment.*

## 3. Extended Kalman Filter for External Wrench Estimation

In this Section, the Extended Kalman Filter (EKF) for interaction wrench estimation is designed. The authors defined an augmented filter state which comprehends translational and rotational components of position and velocities of the robot, respectively $\mathbf{x}$ and $\dot{\mathbf{x}}$, and the external interaction wrench $\mathbf{h}_{ext}$:

$$\mathbf{x}_a = [\dot{\mathbf{x}}, \mathbf{x}, \mathbf{h}_{ext}]^T.\tag{11}$$

The augmented filter state $\mathbf{x_a}$ is then substituted in the interaction dynamics Equation (10) to write the state-space interaction dynamics:

$$\dot{\mathbf{x}}_a = \begin{bmatrix} -\mathbf{M}^{-1}\mathbf{D} & -\mathbf{M}^{-1}\mathbf{K} & -\mathbf{M}^{-1}\overline{\mathbf{L}}(\mathbf{q}) \\ 1 & 0 & 0 \\ 0 & 0 & 0 \end{bmatrix} \mathbf{x}_a + \begin{bmatrix} \mathbf{M}^{-1}\mathbf{K} \\ 0 \\ 0 \end{bmatrix} \mathbf{x}^d = \mathbf{A}_{sp}\mathbf{x}_a + \mathbf{B}_{sp}\mathbf{x}^d,\tag{12}$$

where $\mathbf{A}_{sp}$ is the state-space matrix and $\mathbf{B}_{sp}$ is the input matrix.

To account for the uncertainties in the model, a variable $\boldsymbol{\nu}_a = [\boldsymbol{\nu}_\mathbf{x}, \boldsymbol{\nu}_{\dot{\mathbf{x}}}, \boldsymbol{\nu}_{\mathbf{h}_{ext}}]$ is included in the filter dynamics. The resulting equations represent the filter dynamics:

$$\mathbf{f}(\mathbf{x}_a, \boldsymbol{\nu}_a) = \begin{bmatrix} \ddot{\mathbf{x}} \\ \dot{\mathbf{x}} \\ \dot{\mathbf{h}}_{ext} \end{bmatrix} = \begin{bmatrix} \mathbf{M}^{-1}\left(-\mathbf{D}\dot{\mathbf{x}} - \mathbf{K}\mathbf{x} - \overline{\mathbf{L}}(\mathbf{q})\mathbf{h}_{ext} + \mathbf{K}\mathbf{x}^d + \boldsymbol{\nu}_\mathbf{x}\right) \\ \dot{\mathbf{x}} + \mathbf{M}^{-1}\boldsymbol{\nu}_{\dot{\mathbf{x}}} \\ \boldsymbol{\nu}_{\mathbf{h}_{ext}} \end{bmatrix},\tag{13}$$

Therefore, calling $\widehat{\mathbf{x}}_a$ the augmented state estimate, $\mathbf{C}_a$ the observation matrix for the robot velocity $\dot{\mathbf{x}}$ and the robot position $\mathbf{x}$, and $\mathbf{K}_{EKF}$ the gain matrix, the EKF is defined as:

$$\begin{cases} \dot{\widehat{\mathbf{x}}}_a = \mathbf{f}(\mathbf{x}_a, \boldsymbol{\nu}_a) + \mathbf{K}_{EKF}(\mathbf{y} - \mathbf{C}_a\widehat{\mathbf{x}}_a), \\ \widehat{\mathbf{y}} = \mathbf{h}(\mathbf{x_a}, \mathbf{w}), \end{cases}\tag{14}$$

where the gain matrix $\mathbf{K}_{EKF}$ is computed as follows:

$$\mathbf{K}_{EKF} = \mathbf{P}\mathbf{C}_a\mathbf{R}^{-1}. \tag{15}$$

**R** represents the measurements noise covariance matrix:

$$\mathbf{R} = \mathbf{H}E\{\mathbf{w}\mathbf{w}^T\}\mathbf{H}^T = \mathbf{H}\mathbf{W}\mathbf{H}^T. \tag{16}$$

The observation function **h** linearly maps the sample inaccuracies, due to measurement noise **w**, through the matrix **H**:

$$\mathbf{H} = \left.\frac{\partial \mathbf{h}}{\partial \mathbf{w}}\right|_{\widehat{\mathbf{x}}_a}. \tag{17}$$

The covariance matrix **P** and its rate:

$$\dot{\mathbf{P}} = \mathbf{A}_a\mathbf{P} - \mathbf{P}\mathbf{C}_a^T\mathbf{R}^{-1}\mathbf{C}_a\mathbf{P} + \mathbf{Q} + \mathbf{P}\mathbf{A}_a^T, \tag{18}$$

are based on the dynamics of the state and on the model uncertainties. Matrices $\mathbf{A}_a$ and $\mathbf{G}_a$ are defined, respectively, as:

$$\mathbf{A}_a = \left.\frac{\partial \mathbf{f}}{\partial \mathbf{x}_a}\right|_{\widehat{\mathbf{x}}_a}; \qquad\qquad \mathbf{G}_a = \left.\frac{\partial \mathbf{f}}{\partial \mathbf{v}_a}\right|_{\widehat{\mathbf{x}}_a}. \tag{19}$$

Matrix **Q**, used for the estimation of the parameters, is defined as:

$$\mathbf{Q} = \mathbf{G}_a E\{\mathbf{v}_a\mathbf{v}_a^T\}\mathbf{G}_a^T = \mathbf{G}_a\mathbf{V}\mathbf{G}_a^T. \tag{20}$$

It has to be mentioned that it is possible to neglect the evaluation of the time-derivative $\dot{\overline{\mathbf{L}}}(\mathbf{q})$ in Equation (19). Instead, $\overline{\mathbf{q}} = \mathbf{q}$ is updated as soon as the values of the Jacobian $\mathbf{J}(\mathbf{q})$ and the inertia matrix $\mathbf{B}(\mathbf{q})$ are affected by the modifications of the joint configuration. This assumption is justified by the small change in the joint configuration when the robot interacts with the environment while performing a task, such as assembly, or at least such modification dynamics is much slower than the dynamics of the interaction. Therefore, Equation (13) can be modified accordingly:

$$\mathbf{f}(\mathbf{x}_a, \mathbf{v}_a) = \begin{bmatrix} \ddot{\mathbf{x}} \\ \dot{\mathbf{x}} \\ \dot{\mathbf{h}}_{ext} \end{bmatrix} = \begin{bmatrix} \mathbf{M}^{-1}\left(-\mathbf{D}\dot{\mathbf{x}} - \mathbf{K}\mathbf{x} - \overline{\mathbf{L}}(\overline{\mathbf{q}})\mathbf{h}_{ext} + \mathbf{K}\mathbf{x}^d + \mathbf{v}_\mathbf{x}\right) \\ \dot{\mathbf{x}} + \mathbf{M}^{-1}\mathbf{v}_{\dot{\mathbf{x}}} \\ \mathbf{v}_{\mathbf{h}_{ext}} \end{bmatrix}. \tag{21}$$

**Remark 2.** *The proposed EKF has been discretized for its implementation and online usage [34].*

## 4. Simulation Results

In this Section, the results of the interaction wrench's estimation are evaluated, deploying the proposed Extended Kalman Filter in different simulations. The Robotics Toolbox [35] is used to implement the kinematics and dynamics of Franka EMIKA panda.

The robot is controlled through the sensorless Cartesian impedance control described in Section 2. The impedance control matrices have been imposed as diagonals and the parameters are selects as follows: the mass parameters of the diagonal matrix **M** have been selected equal to 10 kg while the inertia parameters have been imposed equal to 10 kg/m$^2$; the translation and the rotational parameters of the diagonal stiffness matrix **K** have been selected respectively equal to 1000 N/m and 5000 Nm/rad; the diagonal matrix **h** is composed of damping ratio parameters equal to 2. The damping ratio can be exploited in order to compute the damping matrix as $\mathbf{D} = 2\mathbf{h}\sqrt{\mathbf{M}\mathbf{K}}$.

Since the Franka EMIKA panda robot is redundant, its null-space configuration must be managed. In the proposed robot control implementation, a pure damping behavior is exploited for the null-space configuration management, damping the null-space motion:

$$\boldsymbol{\tau}_R = \mathbf{B}(\mathbf{q}) \left( \left( \mathbf{I} - \mathbf{J}(\mathbf{q})^{\#}\mathbf{J}(\mathbf{q}) \right) (-\mathbf{D}_n \dot{\mathbf{q}}) \right), \tag{22}$$

where $\boldsymbol{\tau}_R$ is the null-space control torque, $\mathbf{I}$ is the identity matrix, $\mathbf{J}(\mathbf{q})^{\#}$ is the pseudo-inverse of the Jacobian matrix, and $\mathbf{D}_n$ is the null-space damping diagonal matrix. The term $\left( \mathbf{I} - \mathbf{J}(\mathbf{q})^{\#}\mathbf{J}(\mathbf{q}) \right)$ is the null-space projection matrix. The term $-\mathbf{D}_n\dot{\mathbf{q}}$ allows the dampening of the null-space motion. The control law Equation (2) is, therefore, modified as follows:

$$\boldsymbol{\tau} = \mathbf{B}(\mathbf{q})\gamma + \mathbf{C}(\mathbf{q}, \dot{\mathbf{q}}) + \mathbf{g}(\mathbf{q}) + \boldsymbol{\tau}_f(\dot{\mathbf{q}}) + \boldsymbol{\tau}_R. \tag{23}$$

The control torque $\boldsymbol{\tau}_R$ acts in the null-space of the manipulator, i.e., not affecting the Cartesian motion of the robot. Indeed, the Cartesian controlled robot behavior in [10] is not affected by this term, together with the proposed estimation provided by the EKF in Section 3.

Four simulation scenarios have been implemented: #1 constant external wrench applied to the robot; #2 variable-sinusoidal external wrench applied to the robot; #3 probing task in a full-coupled robot-environment scenario; #4 sliding task on a stiff environment. In the following, such scenarios are analyzed.

### 4.1. #1 Constant External Wrench

A constant external wrench is imposed after 0.5 s from the starting of the simulation, with a magnitude of 20 N for the interaction forces $\mathbf{f}$, and 5 Nm for the interaction torques $\mathbf{C}$. In Figure 1 shows the estimated interaction forces $\widehat{\mathbf{f}}$ and interaction torques $\widehat{\mathbf{C}}$ vs. the applied interaction forces $\mathbf{f}$ and torques $\mathbf{C}$ are represented. In Figure 2, the force estimation error $\widehat{\mathbf{e}}_f$ and the torque estimation error $\widehat{\mathbf{e}}_C$ are shown. As it can be seen from the provided plots, a fast dynamics is achieved (bandwidth of the implemented EKF of 5 Hz, an order of magnitude higher than the one implemented by the sensorless Cartesian impedance controller). A zero steady-state estimation error is achieved. The obtained performance shows the capabilities of the algorithm to perform the estimation in the proposed scenario. The *rms* has been also computed for each force and torque error component to show the limited generalized mean estimation error.

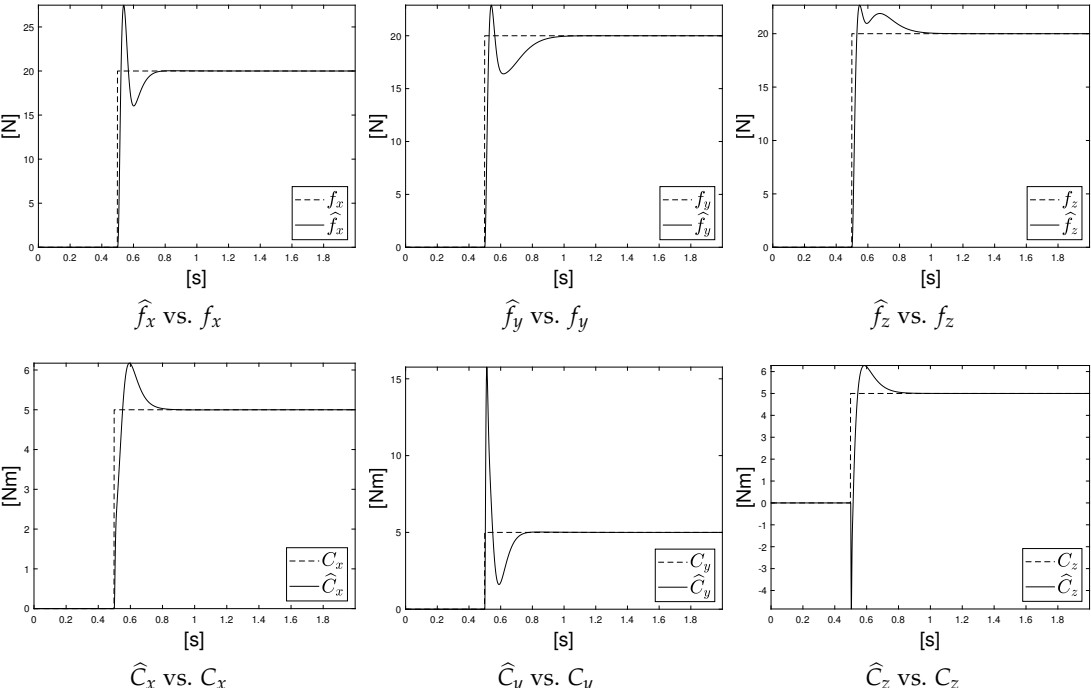

**Figure 1.** Estimated interaction forces $\widehat{\mathbf{f}}$ and torques $\widehat{\mathbf{C}}$ (continuous line) vs. real interaction forces $\mathbf{f}$ and torques $\mathbf{C}$ (dashed line) for the #1 simulation scenario.

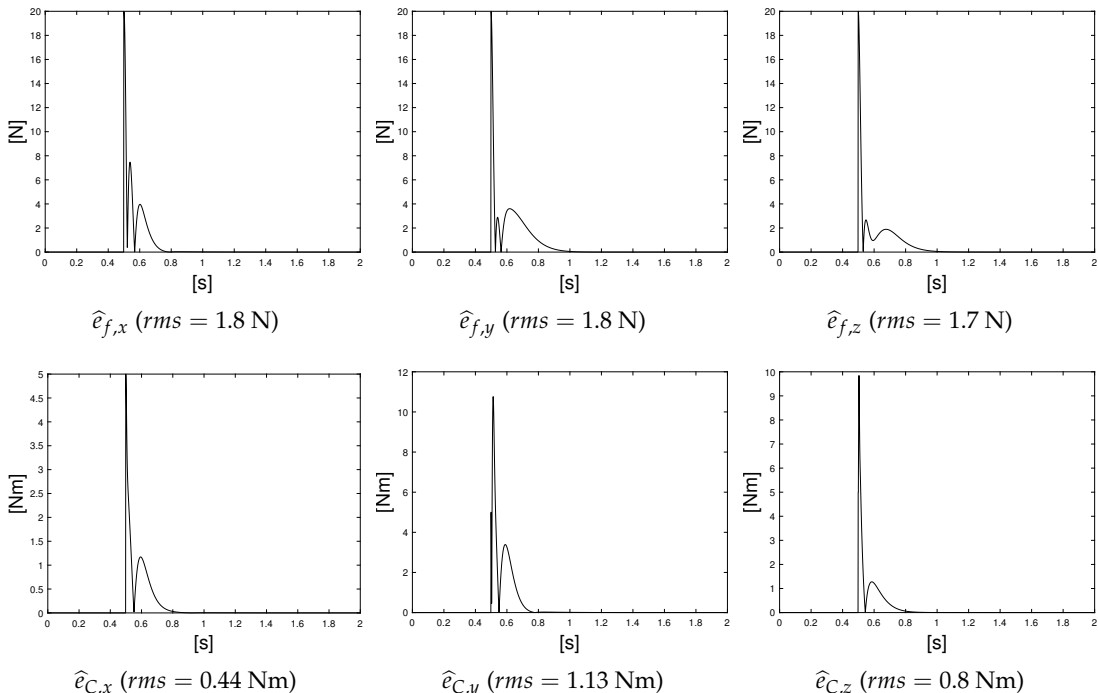

Figure 2. Estimated interaction forces $\widehat{\mathbf{e}}_f$ and torques $\widehat{\mathbf{e}}_C$ errors for the #1 simulation scenario.

### 4.2. #2 Variable-Sinusoidal External Wrench

A variable-sinusoidal excitation is applied to the robot to verify the capabilities of the proposed EKF within a dynamic scenario. The applied external wrench is in the following form:

$$\mathbf{h_{ext}} = \mathbf{A} \cos\left(2\pi\mathbf{f_1}\left(1 + \frac{1}{20}\cos(2\pi\mathbf{f_2}t)\right)t + \phi\right). \tag{24}$$

The diagonal matrix $\mathbf{A}$ contains the magnitudes of the applied forces/torques ($[20, 25, 30]$ N have been considered for the interaction forces $\mathbf{f}$, and $[5, 7.5, 10]$ Nm for the interaction torques $\mathbf{C}$). $\mathbf{f_1}$ and $\mathbf{f_2}$ define the frequencies of the variable-sinusoidal profile, and the related parameters have been imposed to 0.25 Hz for $\mathbf{f_1}$, and 0.85 Hz for $\mathbf{f_2}$. $\phi$ contains the random phases of the variable-sinusoidal profile. In Figure 3 the estimated interaction forces $\widehat{\mathbf{f}}$ and interaction torques $\widehat{\mathbf{C}}$ vs. the applied interaction forces $\mathbf{f}$ and torques $\mathbf{C}$ are represented. In Figure 4, the force estimation error $\widehat{\mathbf{e}}_f$ and the torque estimation error $\widehat{\mathbf{e}}_C$ are shown. As it can be seen from the provided plots, the estimation provided by the proposed EKF is capable of following the profile of the applied wrench. The obtained performance shows the capabilities of the algorithm to perform the estimation in the proposed scenario. The *rms* has been also computed for each force and torque error component to show the limited generalized mean estimation error.

### 4.3. #3 Probing Task

A probing task has been simulated, with the robot approaching a target environment along the $z$ vertical direction. Once the environment is reached, a full-coupled robot-environment scenario is simulated, modeling the target environment as a pure elastic system with a stiffness matrix with the following parameter values: stiffness along translational DoFs: $[10,000, 15,000, 40,000]$ N/m; stiffness along rotational DoFs: $[100, 100, 100]$ Nm/rad. In Figure 5 the estimated interaction forces $\widehat{\mathbf{f}}$ and interaction torques $\widehat{\mathbf{C}}$ vs. the applied interaction forces $\mathbf{f}$ and torques $\mathbf{C}$ are represented. In Figure 6, the force estimation error $\widehat{\mathbf{e}}_f$ and the torque estimation error $\widehat{\mathbf{e}}_C$ are shown. As it can be seen from the provided plots, a fast dynamic is achieved. A zero steady-state estimation error is achieved. Limited transition errors are shown. The obtained performance shows the

capabilities of the algorithm to perform the estimation in the proposed scenario. The *rms* has been also computed for each force and torque error component to show the limited generalized mean estimation error.

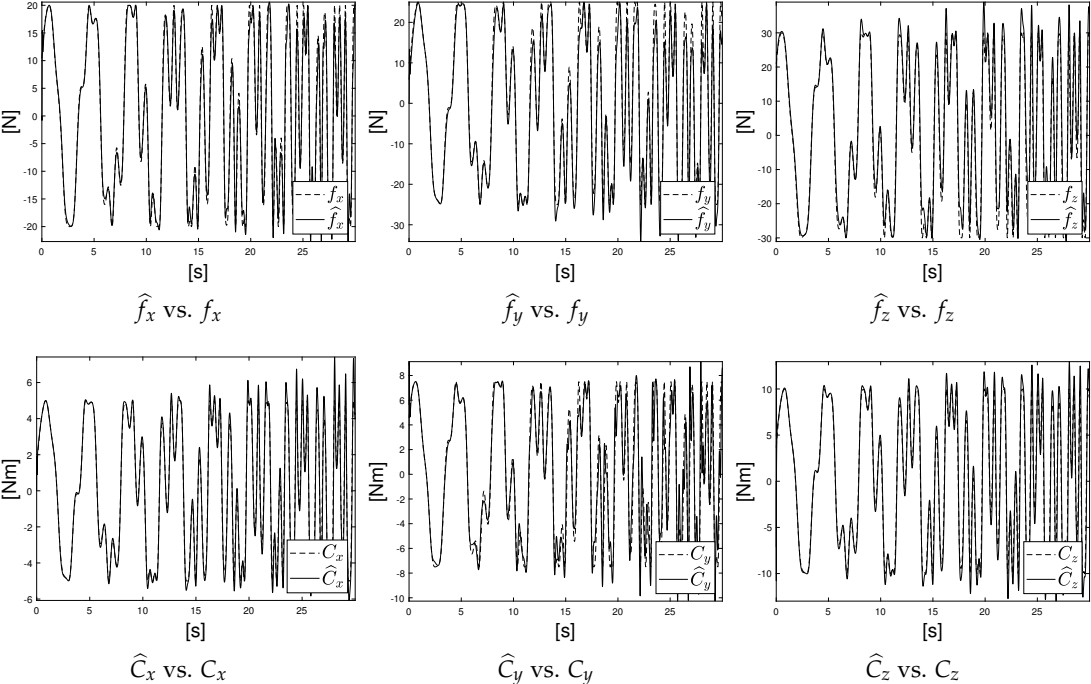

**Figure 3.** Estimated interaction forces $\widehat{\mathbf{f}}$ and torques $\widehat{\mathbf{C}}$ (continuous line) vs. real interaction forces $\mathbf{f}$ and torques $\mathbf{C}$ (dashed line) for the #2 simulation scenario.

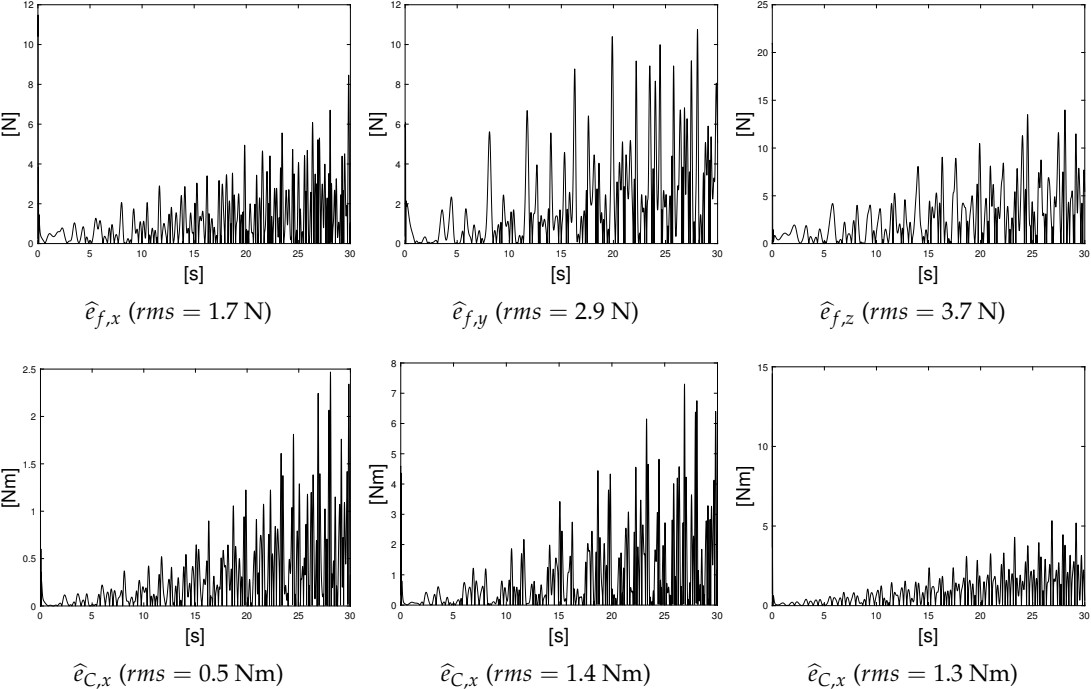

**Figure 4.** Estimated interaction forces $\widehat{\mathbf{e}}_f$ and torques $\widehat{\mathbf{e}}_C$ errors for the #2 simulation scenario.

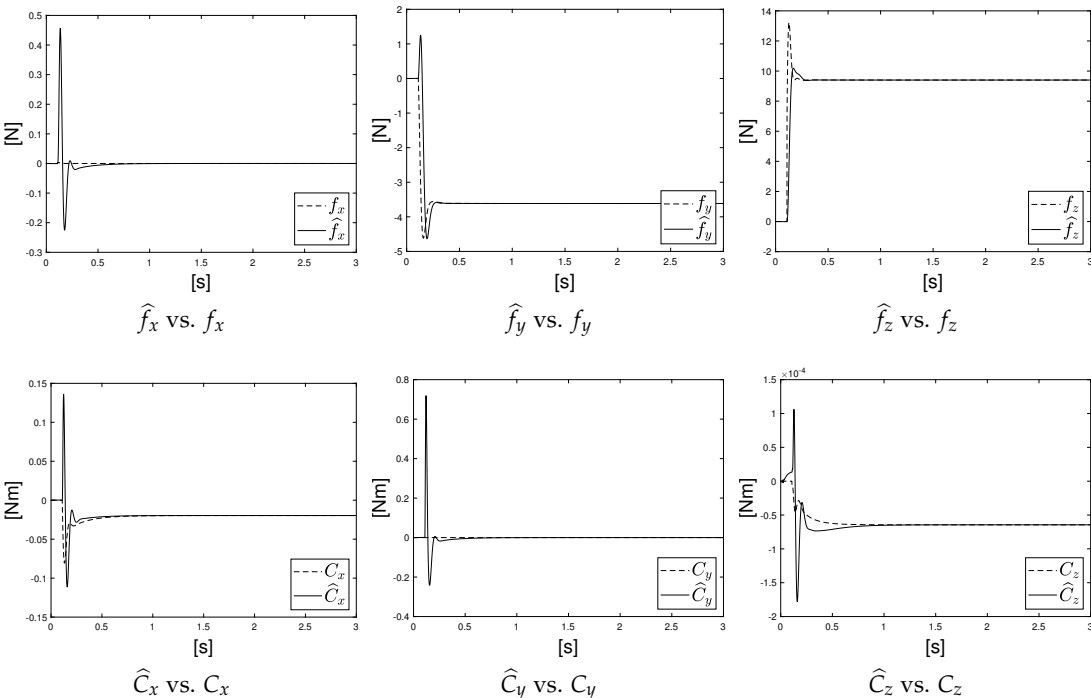

**Figure 5.** Estimated interaction forces $\widehat{\mathbf{f}}$ and torques $\widehat{\mathbf{C}}$ (continuous line) vs. real interaction forces $\mathbf{f}$ and torques $\mathbf{C}$ (dashed line) for the #3 simulation scenario.

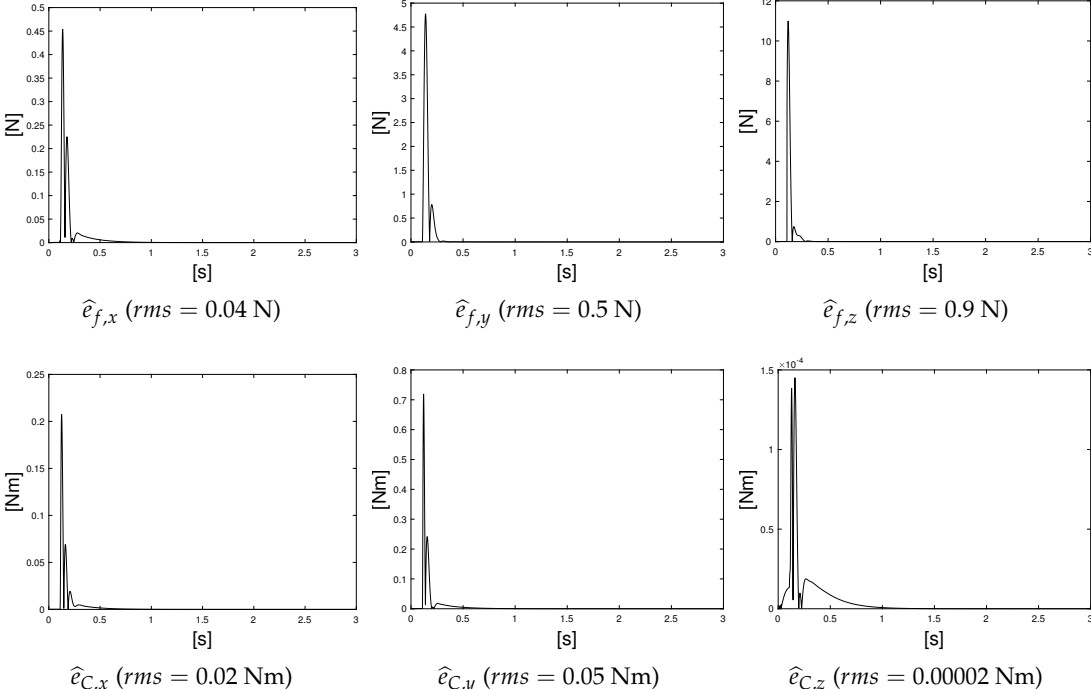

**Figure 6.** Estimated interaction forces $\widehat{\mathbf{e}}_f$ and torques $\widehat{\mathbf{e}}_C$ errors for the #3 simulation scenario.

### 4.4. #4 Sliding Task

A sliding task has been simulated, with the robot in contact along the $z$ vertical direction and sliding on the surface of the environment along $x$ and $y$ directions. Friction forces have been simulated ($f_{f,x} = -\dot{x}_{t,x}F_x$, $f_{f,y} = -\dot{x}_{t,y}F_y$, with $F_x = 30\,\mathrm{Ns/m}$, and $F_y = 50\,\mathrm{Ns/m}$). A stiffness parameter $K_{e,z} = 40{,}000\,\mathrm{N/m}$ has imposed to model the elastic contact between the robot and the environment. In Figure 7 the estimated interaction forces $\widehat{\mathbf{f}}$ and interaction torques $\widehat{\mathbf{C}}$ vs. the applied interaction forces $\mathbf{f}$ and torques $\mathbf{C}$ are

represented. In Figure 8, the force estimation error $\widehat{\mathbf{e}}_f$ and the torque estimation error $\widehat{\mathbf{e}}_C$ are shown. As it can be seen from the provided plots, a fast dynamic is achieved. A zero steady-state estimation error is achieved. Limited transition errors are shown. The obtained performance shows the capabilities of the algorithm to perform the estimation in the proposed scenario. The *rms* has been also computed for each force and torque error component to show the limited generalized mean estimation error.

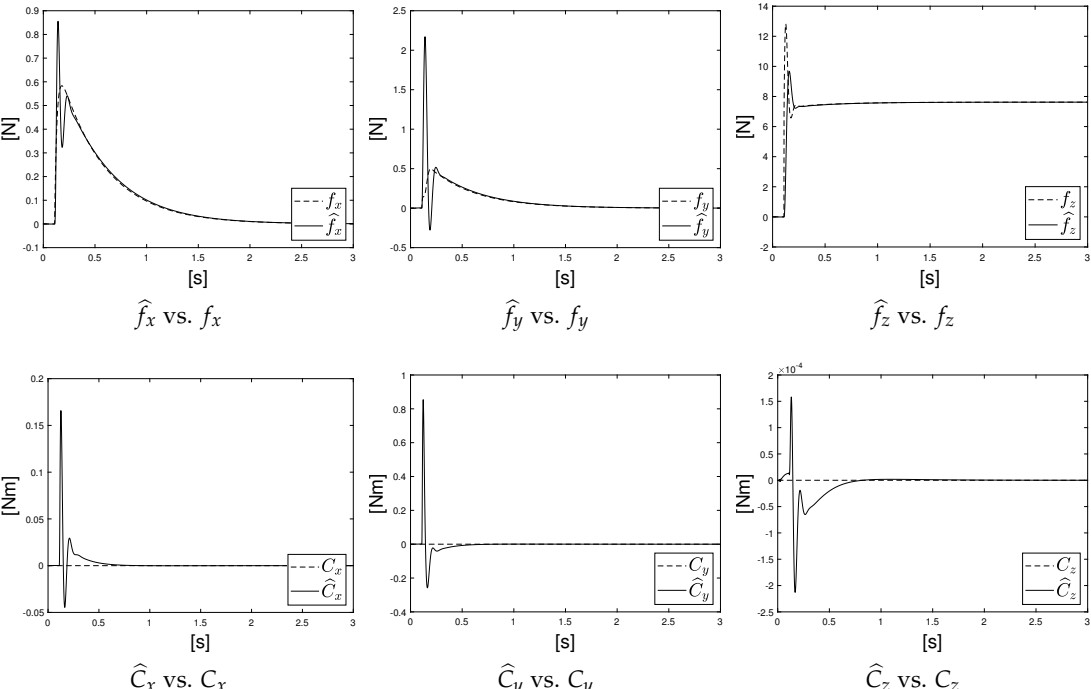

**Figure 7.** Estimated interaction forces $\widehat{\mathbf{f}}$ and torques $\widehat{\mathbf{C}}$ (continuous line) vs. real interaction forces $\mathbf{f}$ and torques $\mathbf{C}$ (dashed line) for the #4 simulation scenario.

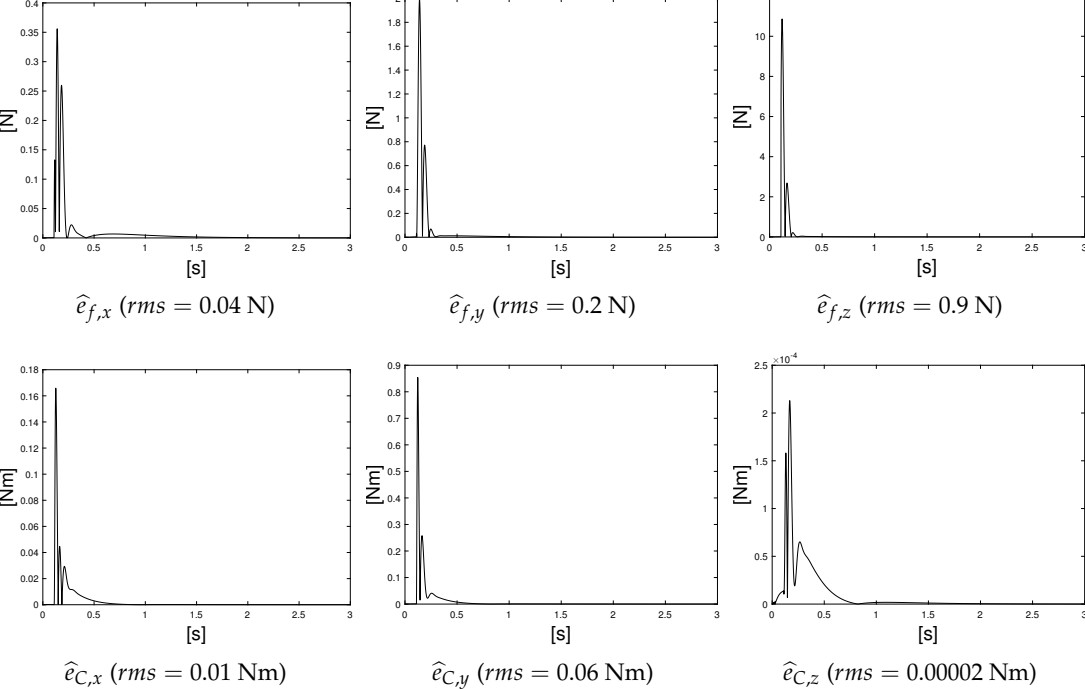

**Figure 8.** Estimated interaction forces $\widehat{\mathbf{e}}_f$ and torques $\widehat{\mathbf{e}}_C$ errors for the #4 simulation scenario.

## 5. Experimental Results

In this Section, experimental results related to the evaluation of the proposed EKF for the estimation of the interaction wrench are shown. A Franka EMIKA panda robot has been employed as a test platform. The wrench estimation provided by the EKF has been compared with the measured wrench obtained from the robot (exploiting its joint-level torque sensors).

The sensorless Cartesian impedance control in Section 2 has been employed to control the robot. The impedance control matrices have been imposed as diagonals and the parameters are selects as follows: the mass parameters of the diagonal matrix **M** have been selected equal to 10 kg while the inertia parameters have been imposed equal to 10 kg m$^2$; the translation and the rotational parameters of the diagonal stiffness matrix **K** have been selected respectively equal to 1000 N/m and 5000 Nm/rad; the diagonal matrix **h** is composed of damping ratio parameters equal to 1.

To manage the redundancy of the robot, the controller in Equation (23) has been implemented. The friction compensation has been performed as proposed in [36].

Two experimental scenarios have been tested: #1 a human–robot interaction scenario; #2 an assembly task. In the following, such scenarios are analyzed.

### 5.1. #1 Human–Robot Interaction

In the here proposed scenario, the robot is controlled exploiting the proposed sensorless Cartesian impedance controller, maintaining a fix setpoint. The human interacts with the manipulator along its kinematic chain, applying forces and torques. In Figure 9 the estimated interaction forces $\widehat{\mathbf{f}}$ and interaction torques $\widehat{\mathbf{C}}$ vs. the applied interaction forces $\mathbf{f}$ and torques $\mathbf{C}$ are represented. In Figure 10, the force estimation error $\widehat{\mathbf{e}}_f$ and the torque estimation error $\widehat{\mathbf{e}}_C$ are shown. As it can be seen from the provided plots, a fast dynamic is achieved. Limited errors are shown during the human–robot interaction. In particular, most of the estimation errors are shown around zero forces/torques. Such estimation errors are related to the non-perfect friction compensation, resulting in fictitious external wrench. The obtained performance shows the capabilities of the algorithm to perform the estimation in the proposed scenario. The *rms* has been also computed for each force and torque error component to show the limited generalized mean estimation error.

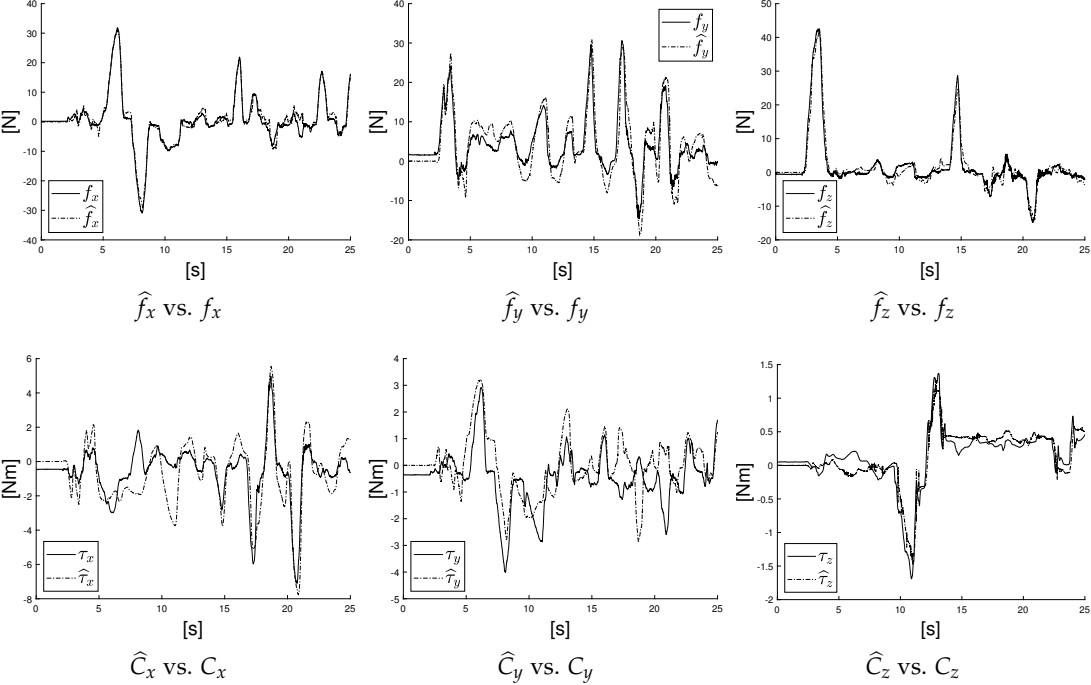

**Figure 9.** Estimated interaction forces $\widehat{\mathbf{f}}$ and torques $\widehat{\mathbf{C}}$ (continuous line) vs. measured interaction forces $\mathbf{f}$ and torques $\mathbf{C}$ (dashed line) for the #1 experimental scenario.

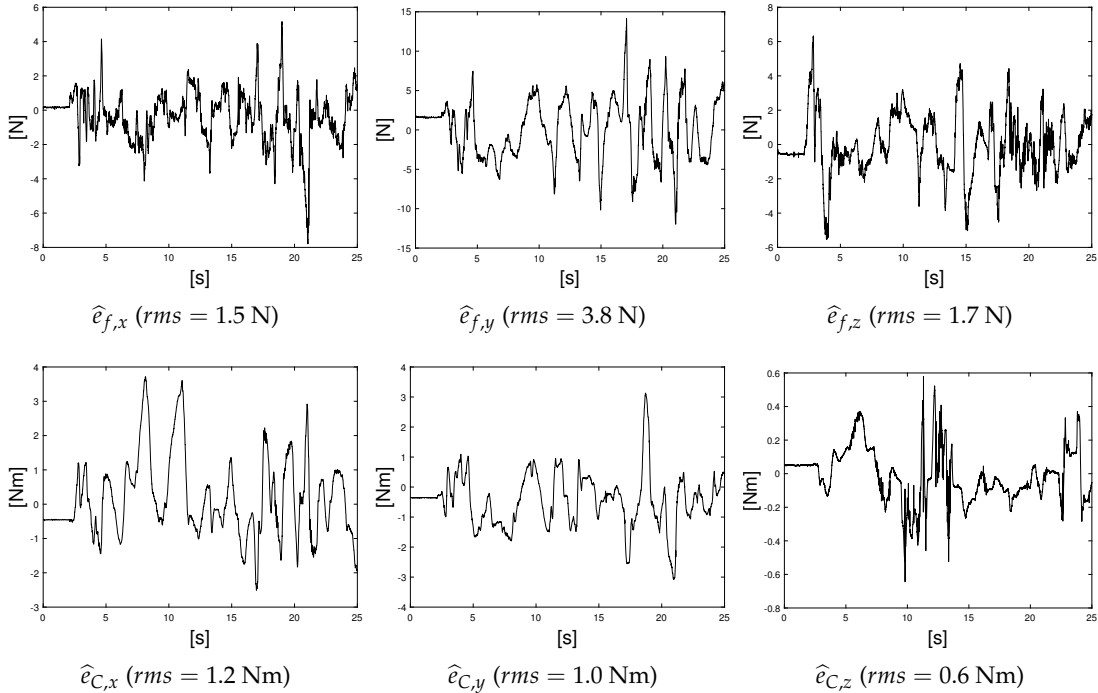

**Figure 10.** Estimated interaction forces $\widehat{\mathbf{e}}_f$ and torques $\widehat{\mathbf{e}}_C$ errors for the #1 experimental scenario.

### 5.2. #2 Assembly Task

The proposed task consists of an assembly of a gear into its shaft. The target task is shown in Figure 11. The main task direction is $z$ and, therefore, a reference force $f_z^d = 30$ N has been defined to perform the insertion task. A PI controller has been implemented in order to track such a reference force, exploiting the estimated force $\widehat{f}_z$ to close the control loop. In Figure 12 the estimated interaction forces $\widehat{\mathbf{f}}$ and interaction torques $\widehat{\mathbf{C}}$ vs. the applied interaction forces $\mathbf{f}$ and torques $\mathbf{C}$ are represented. In Figure 13, the force estimation error $\widehat{\mathbf{e}}_f$ and the torque estimation error $\widehat{\mathbf{e}}_C$ are shown. As it can be seen from the provided plots, a fast dynamic is achieved. A limited steady-state estimation error is achieved (around 3 N for forces, around 0.3 Nm for torques). Limited transition errors are shown. The obtained performance shows the capabilities of the algorithm to perform the estimation in the proposed scenario. In addition, the proposed experiment was able to show the possibility to exploit the wrench estimation for control purposes. The *rms* has been also computed for each force and torque error component to show the limited generalized mean estimation error.

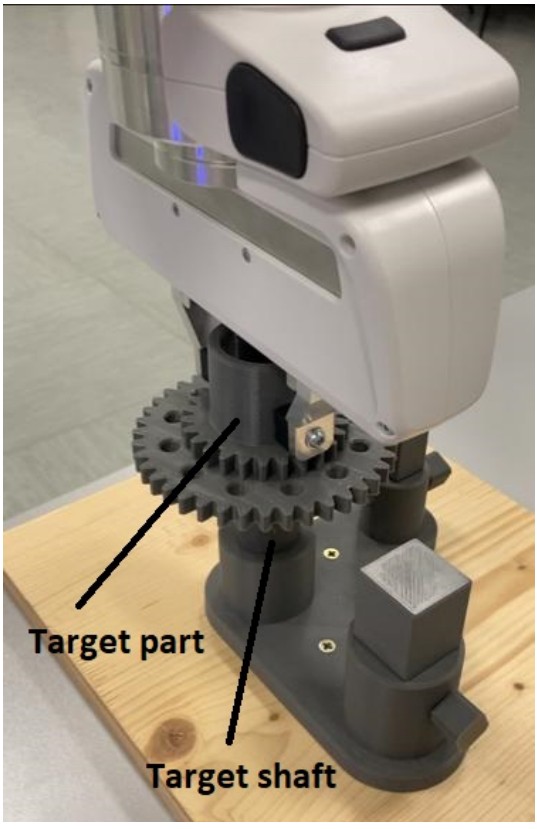

**Figure 11.** Experimental assembly task, including the Franka EMIKA panda manipulator and the target gear to be installed.

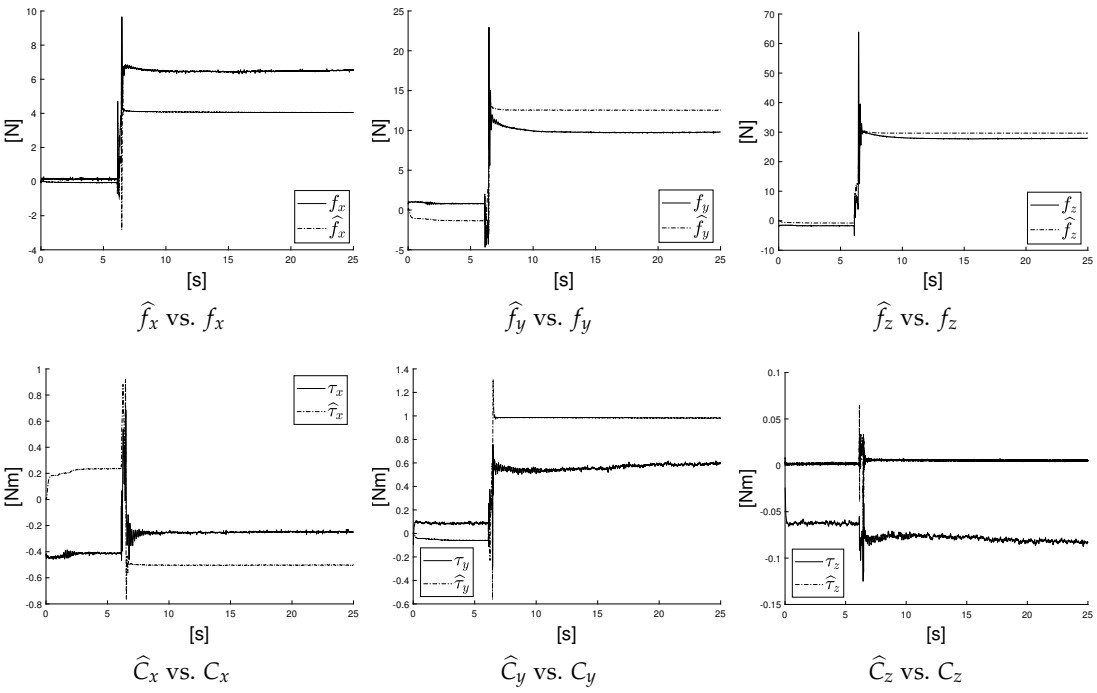

**Figure 12.** Estimated interaction forces $\widehat{\mathbf{f}}$ and torques $\widehat{\mathbf{C}}$ (continuous line) vs. measured interaction forces $\mathbf{f}$ and torques $\mathbf{C}$ (dashed line) for the #2 experimental scenario.

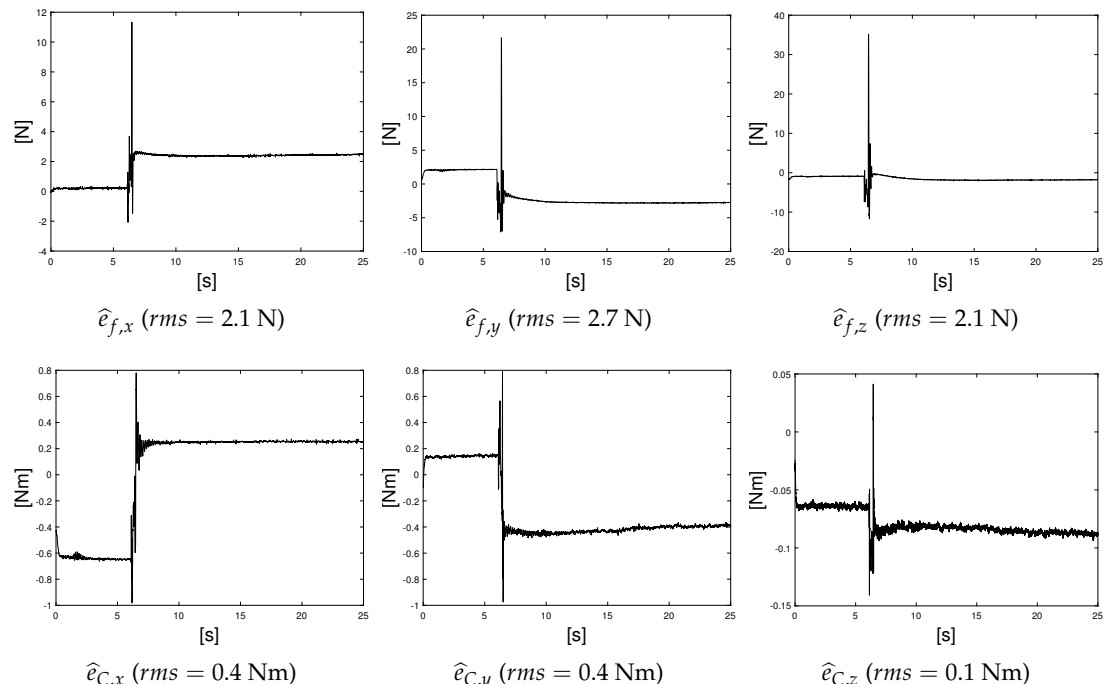

**Figure 13.** Estimated interaction forces $\widehat{\mathbf{e}}_f$ and torques $\widehat{\mathbf{e}}_C$ errors for the #2 experimental scenario.

## 6. Conclusions

The presented paper proposed a sensorless model-based methodology (exploiting sensorless Cartesian impedance control and Extended Kalman Filter) to estimate the interaction wrench. The applied methodology is therefore capable of implementing a 6D virtual sensor for the estimation of both interaction forces and torques. The described approach has been validated in both simulations and experiments, employing a Franka EMIKA panda manipulator. Simulation and experimental results show fast dynamics performing the proposed estimation and limited estimation errors. Estimation errors are shown mostly at zero interaction (i.e., where friction becomes critical) and at transitions (where the proposed filter dynamics is not able to track the real interaction). The proposed filter can therefore be applied to such robotics applications with dynamics slower than the achieved one, where the interaction forces/torques are needed to close a control loop (such as assembly tasks). Current/future work is devoted to improve the estimation accuracy of the proposed EKF developing local high-performance friction compensation algorithms based on learning techniques [37]. The design of a sensorless force control exploiting the proposed 6D virtual sensor is under investigation, exploiting SDRE control [38] for the tuning of both impedance matrices and setpoint. The optimization of the EKF gains is under investigation, making use of machine learning techniques.

**Author Contributions:** Conceptualization, L.R.; Formal analysis, L.R. and A.B.; Funding acquisition, L.R.; Methodology, L.R. and A.B.; Project administration, L.R.; Software, L.R. and A.B.; Supervision, F.B. and D.P.; Validation, L.R. Writing – original draft, L.R. and A.B.; Writing–review and editing, L.R. and A.B.; All authors have read and agreed to the published version of the manuscript.

**Funding:** The work has been developed within the project ASSASSINN, funded from H2020 CleanSky 2 under grant agreement n. 886977.

**Conflicts of Interest:** The authors declare no conflict of interest.

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
