# Peer review of "6D Virtual Sensor for Wrench Estimation in Robotized Interaction Tasks Exploiting Extended Kalman Filter"

_machines, doi:10.3390/machines8040067_

Round 1
Reviewer 1 Report
The author states that they have proposed an EKF algorithm. Compared with the traditional EKF algorithm, what are the theoretical differences and the experimental performance? As a nonlinear algorithm, are unscented Kalman filter and particle filter suitable for this estimation?
Author Response
The authors would like to thank the reviewer for the suggestions. The proposed EKF is a standard EKF w.r.t. the implementation. Its derivation is customized to the target problem.
The following has been included in the revised paper, Section 1.3:
The interaction wrench can be considered as a deterministic variable (i.e., a model of the interaction between the robot and the environment can be derived and exploited for its estimation). While other approaches can be used to model the interaction wrench dynamics (such as sequential Monte Carlo, unscented Kalman filter, and particle filtering methodologies [1,2,3,4,5]), they require the measurement (i.e., samples) of the interaction wrench for the training of the algorithm. In many practical cases, this is not possible or, if a force/torque sensor is available, the sensor is also used online, i.e., not requiring the implementation of an estimation algorithm. The here presented approach, instead, exploiting the well-known robot dynamics modeling, is capable to perform the estimation of the interaction wrench without any use of wrench data for the algorithm training.
[1] N. Chopin, P. E. Jacob, O. Papaspiliopoulos. SMC2: an efficient algorithm for sequential analysis of state-space models. arXiv:1101.1528, 2013.
[2] L. Martino, J. Read, V. Elvira, F. Louzada, Cooperative Parallel Particle Filters for on-Line Model Selection and Applications to Urban Mobility, Digital Signal Processing Vol. 60, pp. 172-185, 2017.
[3] C. Andrieu, A. Doucet, and R. Holenstein. Particle Markov chain Monte Carlo methods. J. R. Statist. Soc. B, 72(3):269–342, 2010.
[4] I. Urteaga, M. F. Bugallo, and P. M. Djuric. Sequential Monte Carlo methods under model uncertainty, IEEE Statistical Signal Processing Workshop (SSP), pages 15, 2016.
[5] Wan, Eric A., and Rudolph Van Der Merwe. "The unscented Kalman filter for nonlinear estimation." Proceedings of the IEEE 2000 Adaptive Systems for Signal Processing, Communications, and Control Symposium (Cat. No. 00EX373). Ieee, 2000.
Reviewer 2 Report
The authors propose an Extended Kalman filtering for tracking of 6 dimensional variables plus an additional variable, the interaction wrench.
From a practical point of view the paper is very interesting. It is also well-written, in general. However, some points must be improved before a possible publication. See my suggestions below.
- First of all, clarify in the introduction and also in Section 2 that interaction wrench is a static variable or a deterministic variable (does not vary with the iteration - time index) - "deterministic'' in the sense that follows a deterministic rule without any random noise perturbation.
- If the interaction wrench is a static variable or a deterministic variable, from a theoretical point view the EKF that you propose is not the optimal solution. For instance, a more correct approach is the following: you should be perform the tracking of the vector x (and its derivative) for a given interaction wrench "h", and then compare different models (i.e., with different interaction wrench "h") according to the marginal likelihood. Other different "h" can be tested maybe generated according to the previous cloud of "h". In this sense, the state-of-the-art discussion must also be improved considering more correct and powerful approaches based on Sequential Monte Carlo and particle filtering, e.g.,
N. Chopin, P. E. Jacob, O. Papaspiliopoulos. SMC2: an efficient algorithm for sequential analysis of state-space models. arXiv:1101.1528, 2013.
L. Martino, J. Read, V. Elvira, F. Louzada, Cooperative Parallel Particle Filters for on-Line Model Selection and Applications to Urban Mobility, Digital Signal Processing Vol. 60, pp. 172-185, 2017.
C. Andrieu, A. Doucet, and R. Holenstein. Particle Markov chain Monte Carlo methods. J. R. Statist. Soc. B, 72(3):269–342, 2010.
I. Urteaga, M. F. Bugallo, and P. M. Djuric. Sequential Monte Carlo methods under model uncertainty, IEEE Statistical Signal Processing Workshop (SSP), pages 15, 2016.
They are particle filtering-based schemes for the joint purpose of tracking dynamical variables and static (or deterministic) parameters, as in your setting. They are also more general and powerful methods. At least a brief discussion can improve substantially the quality of your work.
- Write clearly your state-space model in Section 2, for instance.
- Clarify if you discretize the differential equation and apply the standard, well-known EKF with a discrete time index.
Author Response
The authors would like to thank the reviewer for the provided suggestions.
W.r.t. the first concern, the discussion has been extended in Section 1.3 of the revised paper, stating that the external wrench can be considered a deterministic variable, i.e., making it possible to derive a model for its prediction. In addition, the following discussion has been included:
The interaction wrench can be considered as a deterministic variable (i.e., a model of the interaction between the robot and the environment can be derived and exploited for its estimation). While other approaches can be used to model the interaction wrench dynamics (such as sequential Monte Carlo and particle filtering methodologies [1,2,3,4]), they require the measurement of the interaction wrench for the training of the algorithm. In many practical cases, this is not possible or, if a force/torque sensor is available, the sensor is also used online, i.e., not requiring the implementation of an estimation algorithm. The here presented approach, instead, exploiting the well-known robot dynamics modeling, is capable to perform the estimation of the interaction wrench without any use of wrench data for the algorithm training.
[1] N. Chopin, P. E. Jacob, O. Papaspiliopoulos. SMC2: an efficient algorithm for sequential analysis of state-space models. arXiv:1101.1528, 2013.
[2] L. Martino, J. Read, V. Elvira, F. Louzada, Cooperative Parallel Particle Filters for on-Line Model Selection and Applications to Urban Mobility, Digital Signal Processing Vol. 60, pp. 172-185, 2017.
[3] C. Andrieu, A. Doucet, and R. Holenstein. Particle Markov chain Monte Carlo methods. J. R. Statist. Soc. B, 72(3):269–342, 2010.
[4] I. Urteaga, M. F. Bugallo, and P. M. Djuric. Sequential Monte Carlo methods under model uncertainty, IEEE Statistical Signal Processing Workshop (SSP), pages 15, 2016.
The state-space model has been included in the revised paper, Section 3.
At the end of Section 3, it has been stated with a remark that "the proposed EKF has been discretized for its implementation and online usage [5]".
[5] Roveda, Loris, Niccoló Iannacci, and Lorenzo Molinari Tosatti. "Discrete-time formulation for optimal impact control in interaction tasks." Journal of Intelligent & Robotic Systems 90.3-4 (2018): 407-417.
Reviewer 3 Report
This is too theoretical an article!
The authors wrote a lot of theory, quite a lot of modeling, and little experiment! The authors almost do not analyze the experimental results! All that can be said about the experimental graphs in Fig.5, they are a bit similar to the calculated ones! The article does not contain any numerical estimates of these differences. The authors did not specify the measurement error in the experiment. And this error should be displayed on the charts. Without this it is not clear whether the authors have achieved a positive result
Author Response
The authors would like to thank the reviewer for the suggestions.
Plots showing the estimation force/torque errors have been included in the revised paper to show the achieved performance. Estimation errors are shown mostly at zero interaction (i.e., where friction becomes critical) and at transitions (where the proposed filter dynamics is not able to track the real interaction). The proposed filter can therefore be applied to such robotics applications with dynamics slower than the achieved one, where the interaction forces/torques are needed to close a control loop (such as assembly tasks [1]). In addition, the rms of the estimation errors have been computed for each simulation/experiment to show the limited generalized mean estimation error. These comments have been included in the revised paper.
[1] Roveda, Loris, Niccoló Iannacci, and Lorenzo Molinari Tosatti. "Discrete-time formulation for optimal impact control in interaction tasks." Journal of Intelligent & Robotic Systems 90.3-4 (2018): 407-417.
Round 2
Reviewer 3 Report
The authors took into account all my comments in the article. The article can be published